# Molecular Biosimilarity—An AI-Driven Paradigm Shift

**DOI:** 10.3390/ijms231810690

**Published:** 2022-09-14

**Authors:** Sarfaraz K. Niazi

**Affiliations:** College of Pharmacy, University of Illinois, Chicago, IL 60612, USA; niazi@niazi.com; Tel.: +1-312-297-0000

**Keywords:** biosimilar, analytical assessment, animal testing, clinical pharmacology, clinical efficacy, FDA, EMA, MHRA, WHO

## Abstract

Scientific, technical, and bioinformatics advances have made it possible to establish analytics-based molecular biosimilarity for the approval of biosimilars. If the molecular structure is identical and other product- and process-related attributes are comparable within the testing limits, then a biosimilar candidate will have the same safety and efficacy as its reference product. Classical testing in animals and patients is much less sensitive in terms of identifying clinically meaningful differences, as is reported in the literature. The recent artificial intelligence (AI)-based protein structure prediction model, AlphaFold-2, has confirmed that the primary structure of proteins always determines their 3D structure; thus, we can deduce that a biosimilar with an identical primary structure will have the same efficacy and safety. Further confirmation of the thesis has been established using technologies that are now much more sensitive. For example, mass spectrometry (MS) is thousands of times more sensitive and accurate when compared to any form of biological testing. While regulatory agencies have begun waiving animal testing and, in some cases, clinical efficacy testing, the removal of clinical pharmacology profiling brings with it a dramatic paradigm shift, reducing development costs without compromising safety or efficacy. A list of 160+ products that are ready to enter as biosimilars has been shared. Major actions from regulatory agencies and developers are required to facilitate this paradigm shift.

## 1. Introduction

The term “paradigm shift” was popularized by American physicist and philosopher Thomas Kuhn [1]. A paradigm shift modifies a scientific discipline’s underlying ideas and experimental procedures. Kuhn popularized the term to describe a significant adjustment to a scientific discipline’s fundamental theories and methods. A paradigm shift occurs when there is a crisis with a viable solution. Adopting the solution resolves the crisis, and the definition of a crisis then changes; that is to say, the way of doing things within a paradigm is shifted. This is what is needed today in the field of biosimilars.

Table 1 lists the paradigm shift steps and how they can be instituted to increase the affordability of biosimilars.
ijms-23-10690-t001_Table 1Table 1Paradigm shift steps with regard to biosimilars.Paradigm Shift StepApplication DescriptorWhat is the current paradigm?A biosimilar product is developed using a step-by-step plan starting with an analytical assessment, followed by animal, clinical pharmacology, and efficacy testing (Figure 1).Define the crisisThe cost of developing biosimilars is 100 to 300 million USD each, which is too expensive for many companies.Confirm the crisisThe 130+ biosimilars approved in the EU and US represent only 14 molecules in the EU and 9 in the US out of more than 200+ available options; the cost of approved biosimilars is still too high.Search for solutionsThe regulatory guidelines were examined for rationality, and several redundancies were discovered.Collect solutionsWaive animal, clinical pharmacology, and efficacy testing.Validate solutionsThousands of studies demonstrate that no biosimilar has failed when it is analytically similar to its reference product.Apply solutionsEducate regulatory agencies and developers to adopt a scientific rationale for studies conducted based on analytical assessments alone.Confirm crisis resolutionWhile some agencies have begun revising their guidelines, such as the MHRA, and others allow waivers upon asking, the need for precise directions from agencies remains unfulfilled.The new paradigmWhen in place, biosimilars will be approved if they demonstrate analytical similarity with their reference products within the range of testing variability. Otherwise, they will be rejected as biosimilars and will be allowed to be resubmitted as new drugs.
Figure 1The current paradigm of establishing biosimilarity.
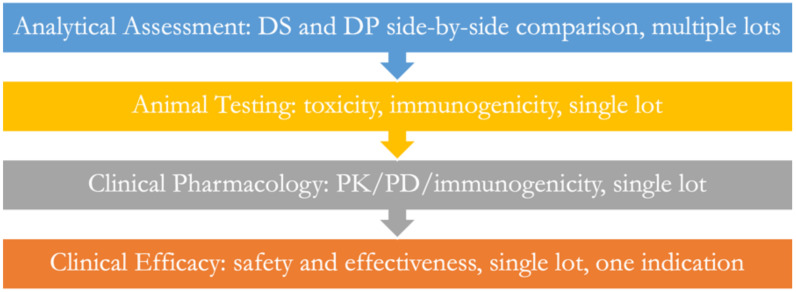


## 2. Understanding Recombinant Proteins

Scientific knowledge about proteins, their recombinant expression, and analytical technologies is continually evolving. Newly developed AI-based bioinformatic models bring solid evidence of 3D structures, enabling the confirmation of biosimilarity without the need for in vivo studies.

The human body makes thousands of proteins by employing just 25,000 genes [2]. The process begins with transcription in the nucleus using DNA as a template to produce pre-mRNA that undergoes post-transcriptional modifications to yield a mature mRNA molecule in eukaryotes. Prokaryotes do not require post-transcriptional changes and can create mature mRNA molecules instantly in the cytoplasm. Ribosomes build polypeptide chains from mRNA template molecules during translation in the cell’s cytoplasm in eukaryotes, where the ribosomes are either free to float or bound to the endoplasmic reticulum. Prokaryotes, which lack a nucleus, carry out transcription and translation within their cytoplasm. Human cells can translate 40,000 to 600,000 proteins [3].

The primary structure of proteins is based on the 20 different naturally occurring amino acids that yield a staggering number of possible proteins, 20^n^ to be exact, where n is the number of amino acid units or residues. Cyrus Levinthal observed in 1969 that an unfolded polypeptide chain has an enormous number of potential conformations as it leaves the ribosome due to the extraordinarily high number of degrees of freedom in the molecule; a predicted value of 10^300^. For instance, a polypeptide with 100 residues will contain 99 peptide bonds and 198 distinct phi and psi bond angles. A protein can thus misfold into a maximum of 3^198^ different conformations if one of three stable conformations can be found for each of these bond angles (including any possible folding redundancy). Therefore, it would take an amino acid chain longer than the universe’s age in years to arrive at its correct native conformation if it achieved its appropriately folded structure by systematically sampling all the potential conformations. Even when conformations are sampled quickly (at nanosecond or picosecond rates), it still takes the same amount of time. The “paradox” is that on a millisecond or even microsecond time scale, the majority of tiny proteins fold on their own. The fast folding also means that there will always be room for error, giving slight differences in the folded protein. For this reason, a protein structure is never a singular structure but a group of equally active structures in their function. Sometimes, the structure variations are large, leading to many diseases treated by injecting silencing RNAs to stop these defective translations.

For almost 50 years, the Cyrus Levinthal paradox maintained uncertainty about the complexity of the 3D structure, which is referred to as the “protein folding problem” [4,5]. A solution to this paradox came in the form of a computational technique that could regularly predict protein structures with atomic precision even when no known structure was identical to it [6]. The technique, found in 2021, validated an entirely redesigned version of the neural network-based model, AlphaFold-2, in the challenging 14th Critical Assessment of protein Structure Prediction (CASP14) [7,8]. The accuracy in most situations proved comparable to experimental structures and far surpassed that of other approaches. Furthermore, the most recent version of AlphaFold-2 is supported by a revolutionary machine learning method that uses multi-sequence alignments to create a deep learning algorithm while incorporating physical and biological facts about proteins’ structures. This research discovery in the field of protein structure prediction was named the “Method of the Year 2021” by Science journal for the unprecedented levels of accuracy achieved by deep learning-based methods when predicting the 3D structures of proteins and protein complexes, essentially solving this long-standing challenge [7].

The code for creating a 3D structure is open source [8]. This approach can be applied to proteins that bind to a target noncovalently, such as mAbs, but also to other proteins such as those that modify covalent bonds, including enzymes, or those that exert activity without particular contacts, including serum albumin, to ensure the similarity of the 3D-derived immunogenicity.

This discovery may aid in the development of biosimilars, which is very encouraging. If the primary structure of a biosimilar candidate is “identical” to the sequence of the reference product, we now know that they both translate to the same 3D structure. Since the tertiary structure is engaged with receptor interaction, we can thus assume that the two products exhibit the same pharmacology and, by extension, the same toxicology. This conclusion further applies to the immunogenicity of proteins that results from their 3D structure.

Since the primary structure can be readily analyzed with confidence, we now have a reason to extrapolate the comparison of the primary sequence to the efficacy and safety of the proposed biosimilar. In addition, the protein structure is no longer a mystery, which it was in the 1970s.

Often, the proteins have terminal amino acids removed in vivo; thus, such variations are acceptable. In many cases, the terminal amino acids are not engaged in structural folding, so these differences are generally allowed. However, we can now check the impact of such differences by running a 3D structure check by repeatedly running the AlphaFold-2 software. For example, if changes in terminal amino acids bring significant changes to the 3D structure, such differences should not be allowed. Repeated modeling is similar to repeated batches, and it demonstrates the randomness of the protein folding process—an observation that was not possible until now. The detailed study of protein structures is generally conducted using X-ray crystallography, nuclear magnetic resonance spectroscopy (NMR), and electron microscopy (EM); while these technologies are now more sensitive, the inherent variability of these testing methods does not allow for total reliance. When combined with AlphaFold-2 predictions, we now have a firm platform to assure that the 3D structure of a biosimilar product is as identical as it can be to the reference product. This approach will reduce the risk of the approval of biosimilars with different primary structures, which is currently allowed.

Other side-effect risks of biosimilars include product- and process-related attributes, which can also be readily analyzed with confidence now, allowing for the establishment of similarity with the reference product.

Recombinant DNA technology was developed in 1972 [9], and at the time, its inventors issued warnings about its use [10] and recommended extensive safety testing. Despite the confirmation of the safety of recombinant drugs, caution remains evident in the testing requirements of new biological drugs, which cost billions to develop, and of biosimilars, which cost hundreds of millions. Despite these reservations by regulatory agencies, the line of biological products grew quickly. Since the discovery of recombinant technology, about 1500 recombinant products have been approved by the US, EU, and Canada [11]. In addition, the FDA has approved over 250 peptides and therapeutic protein products, including monoclonal antibodies, hormones, cytokines, growth factors, enzymes, immunomodulators, and many more [12]. These new therapies have resolved the treatment of many diseases once considered untreatable and created a market of over 300 billion USD for big pharma. However, the cost of the development of new biological drugs and their 12-year exclusivity in the US and 10 years in the EU has kept them out of reach for most patients across the globe.

Recombinant proteins make up the majority of therapeutic proteins currently available on the market. Hundreds are undergoing clinical testing to treat malignancies, immunological disorders, infections, and other diseases. In addition, newly designed proteins are continuously being discovered, such as multi-specific fusion proteins, bispecific mAbs, mAbs coupled with small molecules, and proteins with improved pharmacokinetics.

Therapeutic proteins can be categorized into five groups based on their pharmacological activity: (a) replacing a protein that is lacking or abnormal; (b) enhancing an existing pathway; (c) offering a novel function or activity; (d) interfering with a molecule or organism; and (e) delivering other compounds or proteins, such as a radionuclides, cytotoxic drugs, or effector proteins.

Therapeutic proteins can also be categorized according to their molecular categories, which include enzymes, growth factors, hormones, interferons, interleukins, thrombolytics, blood factors, Fc fusion proteins, anticoagulants, bone morphogenetic proteins, and designed protein scaffolds.

By attaching covalent modifications to polypeptide chains, the cells expand the chemical repertory and information richness of the 20 proteinogenic amino acids. The majority of these protein modifications are attached following the synthesis of the polypeptide chain (translation), so the phrase “post-translational modifications” is frequently used to describe them (PTMs). However, some alterations, such as amino-terminal (N-terminal) protein acetylation or N-glycosylation, occur concurrently with translation [13].

The PTMs that take place on specific sites of the protein are not controlled by the gene that expresses the protein sequence; instead, they are specific to each cellular kind that presents a unique combination of milieu interior such as the presence of enzymes and the thermodynamic conditions during the reaction; it is for this reason that these complex chemical reactions are often not controllable via the alteration of the gene sequence but only by mastering the production conditions during expression.

One PTM is glycosylation, which is distinct from glycans. It is the most frequently used PTM. The terms glycan and polysaccharide are synonyms meaning “compounds consisting of a large number of monosaccharides linked glycosidically”. The carbohydrate component of a glycol conjugate, such as a glycoprotein, glycolipid, or proteoglycan, can also be referred to as a glycan in practice, even if it is merely an oligosaccharide. Monosaccharide O-glycosidic connections make up the majority of glycans.

About half of all proteinogenic amino acids can be modified. The modifications range from simple oligosaccharide structures (2–3 kDa) to complex polypeptide chains (up to 10 kDa), such as the small eukaryotic protein ubiquitin or prokaryotic ubiquitin-like protein, to more complex chemical groups, such as methyl groups (14 Da), acetyl groups (42 Da), or phosphate groups (80 Da) (Pup). The most often changed amino acids are those with side chains with hydroxy, amino, or thiol functional groups (serine, threonine, tyrosine, histidine, aspartate, asparagine, lysine, arginine, and cysteine). Proteins frequently carry many modifications, and certain residues can carry multiple modifications simultaneously (for example, monomethylation, dimethylation, or trimethylation of lysine residues). The fact that some complicated modifications, such as ubiquitin, can undergo changes (such as being phosphorylated) indicates how intricate the underlying regulatory networks are.

PTMs affect the chemical composition of the changed residues and nearby polypeptide sections, affecting the net charge, conformation, binding characteristics, and, ultimately, the protein’s function. The majority of PTMs are dynamic and reversible, meaning that they can be added to or removed from the polypeptide chain by specific enzymes (for example, kinases and phosphatases in the case of phosphorylation, acetyltransferases and deacetylases in the case of acetylation, and ubiquitin ligases and deubiquitinases in the case of ubiquitin). Due to their ability to directly alter the characteristics of their substrates, these enzymes serve as the primary mediators of signal transduction pathways and regulatory mechanisms involving PTMs. However, some PTMs, such as N-terminal acetylation, are irreversible, while others, such as lysine acetylation and S-thiolation, can happen without an enzyme. Furthermore, numerous reactive oxygen, nitrogen, chlorine, and electrophilic species are responsible for causing S-thiolation because they produce various thiol modifications and control particular transcription factors involved in developing detoxification pathways. Therefore, many S-thiolated proteins are involved in the redox control of cellular metabolism; their roles have been discussed elsewhere.

Tens of thousands of alteration sites are often found in large-scale protein modification investigations in eukaryotes.

Most glycosylation processes also take place in bacteria after protein folding. Protein changes in bacteria, however, are significantly less common and more diverse, making it difficult to analyze them. As a result, the amount of starting material needed for modification enrichment procedures applied to bacterial samples is often higher. The chemical nature of bacterial changes frequently necessitates the development of biochemical techniques outside conventional biochemical research. However, the example of protein phosphorylation highlights the tremendous advancements in bacterial PTM research over the past decade. Phosphorylation sites were found in the first global site-specific investigation of the Escherichia coli phosphoproteome, published in 2008 [14]; however, a more recent study found 1883 sites [15]. Advanced biochemical procedures for enriching modified proteins and peptides, their analysis by high-resolution mass spectrometry, and bioinformatic interpretation of the resulting datasets were the driving forces behind this progress. This advancement also resulted in the recent identification of several hundred acid-labile arginine phosphorylation sites in mutants of the Gram-positive model bacteria Bacillus subtilis and Staphylococcus aureus that lack a protein arginine phosphatase (YwlE). The broad microbiology community began evaluating the biological functions of bacterial protein modifications due to similar advancements in the discovery of other bacterial PTMs, particularly lysine acetylation, glycosylation, and pupylation.

Finally, the end of the sugar chain is most often capped by a sialic acid in the form of neuraminic N-acetyl acid (NANA) in human cells; as in many mammals, a part of the sialic acid is in the form of neuraminic N-glycolyl acid (NGNA) because the gene that codes for the enzyme that allows the NANA form to become NGNA is muted and inactive in humans.

In summary, now that the risk of variability in the 3D structure is confirmed based on the primary amino acid sequence, much of the uncertainty surrounding the establishment of analytical similarity is gone. In addition, all other components and attributes of biosimilar candidates can be compared using new technologies that bring greater confidence when establishing biosimilarity.

## 3. Regulatory Evolution

Biosimilars should be about as good a as copy as possible, essentially indistinguishable from the reference product. Each batch of a reference product is a “biosimilar” to its previous batch. When a ribosome translates a protein, it is a “biosimilar” to the previous translation in vivo. This concept of variability is often difficult to grasp. Until the arrival of biosimilars, there were only exact copies of chemical drugs as generic or new drugs that had nothing to be compared with. Thus, when it came to creating guidelines for the approval of biosimilars, agencies took a path that came close to that of developing new biological drugs with few concessions, raising the development cost of bringing them to the market prohibitively [16].

The cautious views of the regulatory agencies were further sensitized by extensive fearmongering by big pharma. They declared that only they knew how to make biological drugs that would be safe. They went as far as petitioning the FDA to bar the FDA from approving biosimilars because the FDA knows their in-process control limits. Suppose the biosimilar developers come up with similar specifications. In that case, the FDA might consider their knowledge of the reference product to approve the biosimilars, essentially claiming that it is unethical for the FDA to approve biosimilars. In one petition by AbbVie, it was claimed that the approval of biosimilars constituted a violation of their 5th Amendment constitutional rights [17]; the petition was denied but left regulatory agencies with a greater sense of caution. It worked, as is evident in the approval of only nine molecules in the US and 14 in the EU out of hundreds of possibilities.

To date, 84 biosimilars have been approved in Europe [18], including hormones, interferons, colony-stimulating factors, antibodies, and necrosis factor inhibitors. In addition, the FDA has approved 38 products [19] containing nine molecules with the same product classes except for parathyroid hormone and FSH; 98 products are approved in India [20], 23 in Latin America [21], 40 in Australia [22], and 26 in Canada [23]. Now that we have an extensive history of the safety and efficacy of biosimilars, a better understanding of the testing methodologies, and a critical realization that the current guidelines must be revised (similar to what happened to the guidelines for generic products), history seems to be repeating itself with the arrival of biosimilars.

Regulatory agencies generally agree on a stepwise approach, as is shown in Figure 1. The analytical assessment is the starting point and the most significant step, followed by animal, clinical pharmacology, and clinical safety and efficacy testing. This order of testing mirrors the development route of new drugs. It is essential to realize that biosimilars are the only class of approved products based on comparative testing with a reference product. Several issues arise in the context of this type of testing, including the relative sensitivity and accuracy of the testing steps, where less sensitive testing comes after the most sensitive testing, negating the purpose of testing.

Animal, clinical pharmacology, and clinical efficacy testing are much less sensitive to clinically meaningful differences, yet they follow the most sensitive analytical assessment. One reason for these tests’ lack of robustness is the use of a single batch of product used for comparison compared to multiple batches used in the analytical assessment. Other misconceptions regarding testing other than the analytical assessment are noted below, which all lead to the conclusion that analytical assessments alone are sufficient to establish biosimilarity.

The EMA was the first to release a biosimilar guideline and product in 2006 [24], and it currently has 95 centrally approved products [25]. The FDA brought in guidelines in 2009 [26] and has 37 products approved [19]. On 14 May 2022 [27], the MHRA issued a final set of guidelines representing the most forward-looking approach that should become ideal guidance for the industry. The details of the regulatory submissions of biosimilars are made public, and 86 European Public Assessment reports (EPAR) were accessible as of April 2022 [28]. The FDA also provides access to the registration filing data [29]. Billions of patients receiving biosimilars have shown that biosimilars are therapeutically equivalent [30], with reports of adverse events being no more frequent than those of the reference product [30], and no biosimilar-specific adverse effects have been added to the product information [31].

## 4. Analytical Assessment

Analytical assessment allows for the comparison of many quality attributes, from structure to post-translational modifications to physicochemical and biological properties. The analytical properties are either product-related (the expression system) or process-related (the manufacturing system), and the methods to test them are well defined and their sensitivity fully established.

The standard methods for testing are listed below:Amino acid sequence. Using high-resolution accurate-mass spectrometry (Orbitrap, QToF) in combination with U(H)PLC technology and amino acid analysis techniques is typical for several methods to determine the amino acid sequence. To provide appropriate peptide fragments for LC-MS/MS, proteins or peptides are typically digested by various enzymes. The amino acid sequence is calculated using the mass of the peptides and the fragmentation data. Amino acid analysis is frequently used in conjunction with these studies to determine the quantitative distribution of the amino acids found in the protein.Mapping peptides. Enzymatic or chemical digestion is used to selectively fragment the chosen protein into distinct peptides, which are then analyzed using high-resolution mass spectrometry (Orbitrap, QToF). Following UPLC-UV (MS) validation, peptide map techniques can be regularly employed for batch release or stability investigations.Sulfhydryl groups and disulfide bridges. Our researchers use high-resolution mass spectrometry (Orbitrap, QToF) and colorimetric tests for free sulfhydryl groups to qualitatively and semi-quantitatively evaluate the position and extent of predicted and mismatched disulfide bridges in molecules where cysteine residues are present.Post-translational changes. To define product acceptability criteria, the strategic approach to PTM analysis is developed in the early stages of product development as part of structural characterization studies, comparability programs, stability studies, or release testing.More advanced structures. Circular dichroism (CD), nuclear magnetic resonance (NMR), FTIR, intrinsic fluorescence investigations, and ultraviolet–visible (UV–vis, second derivative) spectroscopy are used to analyze the higher-order structure. Protein aggregation is investigated using dynamic light scattering, SEC with multi-angle laser light scattering (MALS), sedimentation velocity analytical ultracentrifugation (SV-AUC), and differential scanning calorimetry (DSC).Carbohydrate organization. Although the specifics of glycosylation studies depend on the product under consideration, they frequently entail estimating the amounts of neutral and amino monosaccharides and sialic acids, examining the distribution of glycoforms, and figuring out the glycan structure. Various methods give the requisite structural information, including selective enzymatic cleavage, MALDI-TOF mass spectrometry, HPAEC-PAD, HILIC-FLD, or CE-LIF.Conjugational positions. The location of the toxin’s attachment to the linker and, subsequently, the protein, commonly referred to as the conjugation sites, is identified for antibody–drug conjugates (ADCs). After enzymatic digestion, this is often accomplished using high-resolution mass spectrometry (Orbitrap, Q-tof). Using the same deductive strategies, predicting the site of PEGylation and other conjugated forms is possible.Potency tests or cell-based bioassays. It is common practice to use specialized cell-based bioassays or potency assays, including ELISA, binding assays, competitive assays, cell signaling, ligand binding, proliferation, and proliferation suppression. Functional tests that are investigated should be pertinent to the possible MOA in all therapeutic indications, such as apoptosis, complement-dependent cytotoxicity, antibody-dependent cellular phagocytosis, and antibody-dependent cellular cytotoxicity, among others. However, until there is adequate proof to the contrary, a biological event should be regarded as possibly relevant to the MOA. For instance, functional tests (ADCC, ADCP, and CDC) are unnecessary for a reference product that predominantly targets a soluble antigen.Physiochemical characteristics of proteins. Physical characteristics such as molecular weight using high-resolution mass spectrometry (QToF or Orbitrap), isoform patterns, calculating and validating extinction coefficients, electrophoretic patterns, liquid chromatographic patterns, spectroscopic profiles, and protein quantity are identified using physicochemical characterization programs.Purity and impurities may comprise size-based heterogeneities (aggregates, fragments, and sub-visible/visible particles) and charge-based heterogeneities (acidic and basic variants). Other product alterations are examples of heterogeneities created during the manufacture, handling, and storage of biological products: reduced, oxidized, glycated, misfolded proteins, etc. Aggregation or fragmentation occurs due to the protein unfolding of hydrophobic patches with environmental changes during various stages of the manufacturing process. It might cause immunogenic reactions. Depending on the exposure to different stresses (such as shear, thermal, chemical, freeze–thaw, etc.) and the duration of the exposure, the range of aggregate size spans from soluble aggregates to visible precipitates. During SEC analysis, protein loss brought on by stationary phase interactions and salt-induced aggregation or dissociation is a frequent problem. Hence, sedimentation velocity-analytical ultracentrifugation (SV-AUC), a matrix-free alternative to SEC, is used to measure the size distribution quantitatively.Charge variations are proteo-forms with varying charges that form in different colloidal matrices (such as culture medium, in-process buffers, or formulation) at different stages of the manufacturing process. Therefore, cation exchange (CEX) chromatography of several types is a preferred method. Process-related variants or residuals include cell substrates, e.g., HCPs, HCD, cell culture, and downstream processing residuals. Enzyme-linked immunosorbent assay (ELISA) and real-time or quantitative PCR are the method of choice for HCP and HCD detection and quantitation. These variants are not tested in the drug substance qualification stage, as they are part of the release specification.The stability of the biosimilar candidate should be determined according to ICH Q5C. Stress stability testing is an extension of analytical assessment to demonstrate that the degradation products are similar to the reference product. The pharmacopeia’s general monographs include tests for sterility, endotoxins, microbiological limits, volume in the container, uniformity of dosage units, and permitted particle matter; these tests are release specification tests, and the pharmacopeial standards can be employed. Accelerated and stress stability studies are required to establish degradation profiles and provide a further direct comparison of structural similarity. ICH Q5C and Q1A(R) should be consulted to determine the conditions for stability studies that provide relevant data to be compared.Formulation. To reduce analytical differences, the regulatory authorities require that biosimilar products have the same dose, concentration (or strength), mode of action, and route of administration. Formulations can be different but within the established knowledge of safety risks. Using the same or fewer inactive ingredients in the reference product is preferred. If the formulation contains excipients that have not been used in the formulation of biological products, these should be avoided. It is essential to show that the formulation is appropriate in terms of the active ingredient’s integrity, activity, and strength, as well as its stability, compatibility (i.e., how it interacts with excipients, diluents, and packaging materials), and compatibility. If the primary packaging in contact with the product is different, additional safety studies are required to ensure no unexpected leaching of packaging components into the product. Generally, these studies are difficult to justify; thus, developers are encouraged to use a similar primary packaging material instead. The formulation must not contain excipients that have never been used in a similar product; additionally, all excipients must be free of animal products.

The depth and breadth of research into MS have increased its use for clone selection (e.g., clonal proteome variations, product confirmation, sequencing, glycosylation, and residue alterations), process scaleup (e.g., batch variations, contaminants, and leachates), and clinical investigations, which are all steps in the development of biosimilars (e.g., serum analysis, tissue proteomics, and biomarker discovery). Many improvements in instrumentation have arrived recently, including ionizers such as matrix-assisted laser desorption/ionization, electrospray ionization (ESI), and nano-ESI; mass analyzers such as quadrupole-time of flight (QTOF), triple quadrupole (QqQ), Fourier transform-ion cyclotron resonance (FTICR), orbitrap, and ion-mobility; and fragmentation (specifically, data-independent acquisition (DIA), such as sequential window acquisition of all theoretical spectra (SWATH), and alternating low- and high-energy collision-induced dissociation-MS). Multi-attribute methodology (MAM), liquid chromatography (LC), and capillary electrophoresis coupling techniques can measure many products and process pollutants simultaneously in an automated, high-throughput manner. Modern LC-MS sample preparation and analysis methods provide a superior orthogonal alternative to conventional immunochemical tests for host cell protein identification and quantification at the ppm level. By employing several smaller acquisition windows, DIA techniques such as SWATH overcome the constraints of the restricted low abundant peptides encountered in data-dependent acquisition, expanding the coverage. Sequencing techniques and expanding databases help bioinformatics to deconvolute experimental data. Identifying higher order structures (HOS) employing hydrogen-deuterium exchange and crosslinking chemistries combined with MS is another expanding application in biologic characterization. By clarifying the 3-D conformation dynamics of proteins, these approaches make it possible to anticipate post-translational modifications (PTMs), locations, and the degree of degradation hotspots [32].

Additionally, spectroscopic methods have improved our understanding of structure–function interactions. Modern spectroscopic techniques enable the orthogonal evaluation of secondary and tertiary components. A new technology to confirm secondary and tertiary structure involves spectroscopy such as fluorescence spectroscopy, wherein the samples are tested side-by-side at different temperatures to allow for the differentiation of inter- and intra-molecular bonding [33].

Similar developments have been made in functional characterization for structure–function relationships. Given their high throughput and label-free evaluation techniques, surface plasmon resonance (SPR), biolayer interferometry (BLI), and isothermal titration calorimetry (ITC) continue to be popular for kinetic estimation and quantification. SPR is now a workhorse in applications involving proteins and nucleic acids due to its innovative topologies and solid-state designs. The sensor surfaces have been changed to allow cell-based binding to imitate real-time binding events. Label-free detection, binding, and saturation events are made possible by cell-based binding in SPR and BLI, which considers the expression of receptors on the cell surface. The application of BLI for high throughput formulation screening and stress prediction based on binding profiles are recent developments in protein interaction studies. The potential of ITC has been investigated to enable binding kinetic analysis in addition to standard thermodynamic parameter calculation [34]. Potency assays are a precondition for recommending in vivo research and constitute another degree of functional evaluation. Bioprinting cancer models could be the next big thing in safety and efficacy evaluation that overcomes the difficulties of in vivo preclinical xenograft research [35,36].

Below (Table 2) are the current methodologies used to assess critical quality attributes. Also included are suggested orthogonal methods. Orthogonal testing is not repeat testing using another method; it is testing to examine the same attribute from a different angle. As mentioned above, with higher confidence in the 3D structure of proteins based on their primary sequence, the most uncertain testing is now manageable. All other tests can be conducted with suitable or validated methods, allowing the developers to present a robust analytical assessment profile to regulatory agencies.

There are also a few disagreements regarding how the analytical data should be compared. The FDA issued a set of guidelines to apply statistical modeling to analytical data [37] that caused severe concerns about its rationale. As a result, the guidance was withdrawn [38] and replaced with a new set of guidelines [39] in response to a citizen petition [40]. The new guidelines eliminated the tier 1 assessment of quality attributes as it required arbitrary equivalence criteria of 1.5 × SD of the reference product to define the 90% confidence limit of the biosimilar candidate. Instead, the new recommendation recommends a more realistic and scientifically sound range.

The number of reference product batches required varies with the expected variability. For example, three batches would be sufficient to confirm a higher order structure. Analytical assessments can use development lots, but they must include at least one at-scale cGMP lot used for clinical testing; additionally, the regulatory filing will require a bridging study with at least three PPQ lots. For others, where statistical analysis is conducted, more batches are required. A visual comparison is sufficient for test results presented in printed outputs, such as spectra. For quantitative statistics, data from roughly ten batches of data are required, and the 3Sigma range, which is determined for the reference sample as (ref-3ref, ref + 3ref), offers the most precise conclusion. If the test sample’s MinMax range falls inside the 3Sigma range, the 3Sigma test is valid. A more practical compromise between error rates and the sample size is offered by the 3Sigma method.

While the statistical treatment of data is well supported in the case of analytical assessment as multiple lots of the reference and the test products are used to compare whether the variability of the reference product is the same as that of the biosimilar candidate, this is not the case for all other in vivo tests, including animal, clinical pharmacology, and efficacy testing [41].

## 5. Animal Testing

Animal testing is routine to establish the safety of new drugs. This works well for chemical drugs as the reactive chemical groups interact with many tissues, generating adverse reactions. The toxicity of biological medications is an extension of their pharmacological properties, resulting in receptor binding that is often not possible in animal species [42].

While there is widespread awareness of the ineffectiveness of testing biosimilars on animals, this issue cannot be resolved in the US as it is spelled out in the BPCIA, necessitating a legislative change [26].

Besides being wasteful, animal studies can be harmful if they are used to justify any analytical differences. Even though agencies have begun to soften their tone regarding the need for these studies, none of them have come out to stop this testing, despite the renunciation of such studies for new biological drugs by the US FDA [43,44]. To prevent animal abuse and the risk of approving unsafe biosimilars, the author has suggested banning such studies [45], not just discouraging their use. The developers are advised to challenge their regulatory agencies about the need for such studies. To date, only one agency, the MHRA, has made it clear that, in most situations, a comparative efficacy trial may not be necessary if sound scientific justification supports this course of action [29].

Conclusively, no biosimilar product should be tested on animals. The developers should not follow the example set in the regulatory submissions of biosimilars. In some cases, dozens of animal toxicology studies were submitted, and the FDA refused to review them for lack of relevance [46,47].

Still, suppose arguments are presented in favor of animal testing. In that case, these can be defended based on the structural similarity demonstrated with a further discussion of the lack of extrapolation of immunogenicity and other toxic effects on humans.

## 6. Clinical Pharmacology

Clinical pharmacology studies comprise pharmacokinetic, pharmacodynamic, and immunogenicity testing. While both the FDA and EMA have approved biosimilars without clinical efficacy studies, the suggestion of eliminating clinical pharmacology studies is new. It has never been discussed in the literature or by regulatory agencies. In all likelihood, this suggestion will be met with great resistance. For this reason, this paper presents a detailed argument to justify waiving all clinical testing, including clinical pharmacology profiling.

To make a strong case, we need to examine the history of clinical pharmacology profiling to establish the bioequivalence of generic chemical products. Studies with biosimilars are similar to those with generic chemical products.

When generic chemical drugs arrived in 1984 [48], they were only available for branded drugs that had undergone extensive testing for safety and efficacy. A copy of these drugs with the same chemical structure enabled waivers of efficacy testing (except in a few circumstances where bioequivalence cannot be established). The premise that allowed this waiver was that if the chemical structure is the same, then the pharmacological and toxicological response should also be the same. However, to make sure that the total exposure of the body is the same, generic drugs were required to demonstrate clinical bioequivalence, mostly in healthy subjects. A few key misconceptions in this paradigm need to be identified.

Bioequivalence is defined in the US 21 CFR 320.1 as “the absence of a significant difference in the rate and extent to which the active ingredient or active moiety in pharmaceutical equivalents or pharmaceutical alternatives becomes available at the *site of drug action* [emphasis added] when administered at the same molar dose under similar conditions in an appropriately designed study”. Since the site of action is rarely accessible and, in most cases, unknown (especially for chemical drugs), a surrogate test of blood concentration was proposed with the faulty assumption that the concentration at the site of action will be proportional to the blood concentration.

Like any comparative study, the design must present acceptance criteria of acceptable differences. The acceptance criteria for bioequivalence were based on the alpha value of 0.05, the beta value of 0.80, and allowable variation of 80–125%, set arbitrarily. The 80–125% range comes from keeping the 100% at 20% below the maximum of 125% and keeping the lower end at 20% below the 100% value; this also translated to −0.223 as the natural log of 0.8 and +0.223 as the natural log of 1.25. For narrow therapeutic range drugs, the 90–111.1% range came to give a value of −0.105 for 90% and +0.104 for the upper range. Over time, these “rules” of bioequivalence testing became the golden rules until Dr. Gordon L Amidon, a University of Michigan scientist, suggested in 1995 that drugs with high solubility and high permeability need not be tested for bioequivalence [48]. The FDA retained Dr. Amidon as an advisor and worked for several years until, in 2000, 16 years later, they agreed to allow bioequivalence waivers [49] for the drugs that qualified for the Biopharmaceutical Classification System (BCS); so far, the FDA has issued 14 regulatory guidelines on bioavailability and bioequivalence testing [50].

While some misconceptions regarding bioequivalence testing were removed, many more fundamental issues remained. The BCS basis for bioequivalence waivers should have brought an end to most bioequivalence testing since the reason for the variability lies in the formulation differences only; if a product can release an equivalent amount of monomolecular form of the drug at the site of administration, then one should not expect differences in the PK profile. However, relying on blood concentration to validate the equivalence of drug release at the administration site adds substantial biological variability to the monitored parameters. Comparing the release characteristics directly rather than across a biological barrier is more sensitive. Grounded in the philosophy of simulating physiological conditions to test the drug release profile, the common USP methods and other dissolution and release testing were conducted at 37 °C. This physiological temperature makes the testing less sensitive due to increased solubility at this temperature. As a result, many examples of a lack of correlation between in vitro and in vivo testing kept the bioequivalence testing in place.

The in vitro release testing can be made more sensitive when designed to demonstrate the thermodynamic profile of the product [51], meaning that the release in vitro is tested at lower temperatures when the differences in the intra- and inter-molecular bonding can be made more evident. Other approaches to changing the thermodynamic environment include changing the dielectric constant, using surfactants, and several other changes to the dissolution media. The author attempted to convince the FDA [52]. The response from Dr. Janet Woodcock stated that “the current system seems to work well and does not need a major change but to allow waivers based on novel testing methods, as proposed”. She said “therefore, your Petition is granted in part to the extent that it asks us to open comment on novel dissolution tests that can be used to establish bioequivalence [52]”.

The argument takes a different direction when it comes to parenteral biological drugs. No bioequivalence testing is required when drugs are administered through these routes; for example, when given by the intravenous route, complete bioequivalence is assumed. Other parenteral routes may show non-instantaneous input, but being in a liquid form, there is no reason to assume that a biosimilar product will demonstrate a different absorption profile, even when administered in a different formulation; this argument applies because, in a solution form, the drug is available in a monomolecular dispersion. This should apply even if the administered drug precipitates upon subcutaneous delivery because the solubility of the free drug will be the same.

Another argument against clinical pharmacology studies comes from the realization that these studies are conducted using only one batch, assuming that this batch would represent all other batches. Still, when multiple batches have proven highly similar to the reference product in analytical assessment, it would be fair to assume that clinical pharmacology profiling will not show any difference.

The PK/PD studies are universally conducted for biosimilars; the data reported in clinicaltrials.gov for phase 1 and 2 studies list 370 such studies [53], and none failed the PK/PD profile comparison. There have been rare failures, but these were attributed to study design failures, and repeated studies removed the non-conformity [54,55]. No biosimilar product has been rejected due to clinical pharmacology variability by the FDA or EMA.

Several studies have suggested that the complex system of subcutaneous administration [56] poses the risk of bioavailability of biosimilar products. However, if the formulation is the same, this argument does not hold; even if a formulation has minor differences, the complexity of absorption is related to the molecular weight of the active molecule. Thus, there is no reason to believe that subcutaneous administration will require PK profiling; the complexity of the absorption of therapeutic proteins should apply equally to both the reference and the biosimilar candidate, as is shown in all reported studies on biosimilars [57]. Furthermore, for products administered intravitreally [58], no PK studies can be of value as intravitreal space does not allow for the passage of the drug into the circulation, and it has no immunogenicity receptors. These are some of the identified misconceptions in the current practice of testing biosimilars, such as those shown for aflibercept [59] and ranibizumab [59].

Moving forward past absorption, the reported volume of distribution following IV treatment for mAbs and other significant therapeutic proteins is near the plasma volume, indicating limited transport into tissues [60]. Thus, structural differences that might alter the distribution cannot cause a change in the disposition kinetics.

Several mechanisms are used to remove therapeutic proteins from the bloodstream or interstitial fluid, including proteolytic degradation, target-mediated clearance, Fc receptor-mediated clearance, nonspecific endocytosis, and immune complex formation followed by complement- or Fc receptor-mediated clearance mechanisms. Although the body undergoes a great deal of proteolysis, little is known about its kinetics and mechanics, especially in the case of large therapeutic proteins such as mAbs [61]. However, there is no reason to believe that the clearance of biosimilars will be different if there are no differences in the molecular structure.

Protein degradation byproducts and low molecular weight (MW) biologics (MW 30 kDa) are greatly eliminated by renal excretion. In addition, low MW proteins are widely understood to be filtered, transported, and metabolically processed in the kidneys [62]. Therefore, the same argument presented for clearance also applies here. No study has demonstrated a difference in the disposition profile between biosimilars and their reference products.

In summary, it is unnecessary to conduct PK studies to characterize the disposition profile if the structural similarity has been established. It would be incorrect to assume that, despite the product- and process-related attribute similarity, the disposition profiles or pharmacodynamic responses will differ.

Another important reason not to conduct these studies comes from ethical considerations. When healthy subjects are used in such studies, we inadvertently expose them to an immune response that may stay with them for the rest of their lives and hinder future use of these biological therapies. This risk is not justified based on the arguments presented above.

One argument in favor of PK studies is that they allow pharmacodynamic monitoring; the argument is that, if the structure is highly similar, it is not rational to expect a different pharmacodynamic response. The published data confirm this suggestion. The section on analytical assessment described the several attribute variations responsible for variation in the pharmacodynamic properties; these are better identified at the analytical assessment level than through in vivo testing. The FDA and EMA are now allowing waivers for clinical efficacy testing if the pharmacodynamic profile shows similarity for such compounds as cytokines. A different pharmacodynamic response should not be anticipated when the molecular weight, the primary sequence, and the secondary and tertiary structure are similar to the reference product.

## 7. Immunogenicity

Proteins have the potential to trigger an undesirable immune response, resulting in the stimulation of the formation of anti-drug antibodies. Therefore, as part of the development pathway, comparative clinical immunogenicity testing in an adequately sensitive study population (i.e., the patient population in which the study biologics are most likely to elicit an immune response) is recommended by the EMA, WHO, and FDA as a key criterion for the regulatory evaluation of biosimilarity [63]. Testing details further require a fully validated, tiered approach for detecting ADAs, including testing at several stages, ADA confirmation assays, ADA characterization, titration, and neutralizing capacity assessment [64]. The immunogenicity data highly depend on the assay used to measure ADAs (including reagents, standards, validation criteria, etc.), making this testing highly variable. The most widely used ADA detection methods are bridging enzyme-linked immunosorbent assay (ELISA, which uses labeled therapeutic mAbs) and radioimmunoassay (RIA). However, other new methods, including competitive displacement and tandem mass spectrometry, have also been proposed [63].

In vitro tools for predicting immunogenicity are being developed that may be useful in the future to analyze variations [64]. In addition, the EMA provides recommendations for such predictive tools as supportive information for biosimilarity.

The FDA Biosimilar Action Plan [64] advises using in silico methods to compare biosimilars, such as immunogenicity tests. Better analytical assessment approaches increase our confidence in decreasing or eliminating anti-drug antibody testing because immunogenicity is purely structure dependent. Additionally, as part of the analytical evaluation, extrinsic immunogenicity caused by contaminants and aggregates can be easily quantified and evaluated compared to a reference product.

Finally, the FDA has stated in its recent advice on insulins that, if the immunogenicity profile varies but has no bearing on the disposition profile, the variations are irrelevant [65,66].

As analytical assessments have become more robust and immunogenic reactions are strictly structure-dependent, the need for such testing is less relevant. The current approach to monitoring plasma antibody levels is highly inaccurate and variable; the reliance should move to analytical similarity rather than a secondary response. If analytical assessment assures a similar 3D structure and comparable product- and process-related attributes, the testing for immunogenicity becomes redundant. There is a significant risk in testing as test populations might be sensitive to the drug, causing irreversible damage to their immune systems.

## 8. Clinical Efficacy

The clinical efficacy testing of new drugs against a placebo is a gold standard that has come under criticism recently. Dr. Janet Woodcock, a past acting commissioner of the FDA, asked “why should we put patients through all these different trials just to check a box?” In addition, the FDA is questioning real-time testing and has stated that the clinical efficacy protocols are “broken” [67] and that new digital technologies and real-world evidence (RWE) are required, as outlined in the 21st Century Cure Act [68].

When testing two products that have already been proven highly similar in analytical assessment, this testing becomes least sensitive to identify any clinically meaningful difference. Additionally, the logic of subjecting biosimilar candidates to only one efficacy study, while there may be multiple indications given to the biosimilar product, makes such testing no more than “checking a box”, to paraphrase Dr. Woodcock.

According to an analysis of the published literature, the clinical effectiveness investigations have not revealed any “clinically meaningful difference”, to use the FDA’s terminology, between a biosimilar and its reference product. As a result, they have not led to any market withdrawals or recalls. None of these regulatory filings failed in clinical efficacy testing. The studies reported on the clinicaltrials.gov portal [69] show that all 141 studies for which the results are reported met the acceptance criteria. In addition, the PubMed database lists 435 randomized control clinical trials from 2002 to 2022 that showed no clinically meaningful difference [70]. The same holds true of the safety and efficacy reports available in the FDA’s Adverse Event Reporting System Database [71] and the European EudraVigilance [72]. A 2019 meta-analysis [73] showed that the 38 biosimilars met the comparative clinical efficacy standards.

The clinical efficacy studies never fail, as is evidenced by the European Public Assessment Report (EPAR) [28] and the FDA Approved Drugs Database [29]. The data reported on clinicaltrials.gov show hundreds of entries of biosimilar testing, and all studies that posted the results showed no study failures. The reasons why clinical efficacy testing does not fail are myriad. First, these studies have very low power since the difference anticipated is almost zero. Doing it correctly requires 10 to 100 times more patients than the population used to approve the reference product [46]. Since such studies are impossible to conduct, the study subjects are reduced, leading to study powers at which the study becomes useless and demonstrating that none of these studies failed. Second, as a matter of statistical consideration, an acceptable difference must be specified a priori; there is no rationale, and it is only a clinical judgment. Studies have used anywhere from 2% to 65% difference as acceptable. Third, as required, a single study in one indication cannot assure similar results in another indication, despite the similarity of the mode of action. Fourth, the study endpoints are difficult to establish, particularly when the patients have received prior therapy—it increases inter-patient variability that cannot be assessed a priori to the study’s design. All these combined, along with extensive data showing no efficacy testing failures during the development or after marketing the product, only point to one reality. These studies should not be conducted. If a regulatory agency requires efficacy testing, the developer must ask two questions:Is there a residual uncertainty that requires further clinical testing?Would the proposed study remove such uncertainties for all extrapolated indications?

The FDA has admitted that biosimilars “may be approved based on PK and PD biomarker data without a comparative clinical study with efficacy endpoints” [74], since these studies are more sensitive than the clinical efficacy testing, as is demonstrated in the comparison of the testing of clinical efficacy with endpoint(s) for GCSF [75] when the FDA approved filgrastim-aafi, filgrastim-sndz, filgrastim-ayow (author’s), pegfilgrastim-jmdb, pegfilgrastim-cbqv, pegfilgrastim-pbbk (author’s), and epoetin alfa-epbx. The FDA now fully acknowledges the role of PD markers in establishing biosimilarity.

Another issue that pertains to biosimilars only in the US is that there are two classes of biosimilars, one that has no clinically meaningful difference from the reference product and the other when the same product is subjected to extensive switching and alternating with the reference product to assure that doing so does not reduce the efficacy or increase the adverse effects. It is labeled as an interchangeable biosimilar. A serious objection to this classification [76] teaches that there is no rationale for such differentiation, since none of these studies can ever fail, reducing the exercise to merely a checklist entry costing hundreds of millions of dollars and allowing big pharma to declare an interchangeable biosimilar as superior to other biosimilars. A recent study presented an analysis of about 1000 publications analyzing switching between biosimilars and found no difference in safety or effectiveness [77]. Understanding the molecular basis of biosimilarity teaches us that if two molecules are similar, meaning no different than the variability of the reference product, expecting any other outcome of safety or efficacy is not rational. The issue becomes more controversial when the studies conducted to prove what is not provable are designed so that they cannot prove anything, except concluding that they are similar due to the abovementioned reasons.

More details are available at the testimony of the author to the FDA [78] that includes many suggestions on the design of analytical assessment, clinical pharmacology, nonclinical and efficacy studies.

## 9. New Paradigm

The customary development plan (Figure 1) needs a major change, removing all boxes except the analytical assessment. The science of analytical testing has progressed to a point where we can now clearly point out even the most minor differences in molecular structures. As long as the differences are within the range of differences found in the reference product, these should be of no concern. However, a dichotomy arises when conducting studies where only a single lot of biosimilar and a single lot of the reference product are used in all other studies except the analytical assessment, where multiple lots, as many as ten, are tested to confirm the similarity.

A more straightforward understanding of the sensitivity and reproducibility of testing also teaches us that animal testing, clinical pharmacology profiling, and efficacy testing are much less capable of identifying any difference because there is not supposed to be a difference—it is not the same in the case of placebo-controlled studies where only one arm is supposed to demonstrate a response. When the two products are supposed to be similar, the study power can only be attained when we use hundreds of times the test population, and even then, the differences cannot be established. As a result, all these studies that do not fail because they cannot fail are conducted only as a box-ticking exercise, primarily due to an abundance of caution that is neither necessary nor realistic.

## 10. Available Choices

Table 3 shows the patent expiry of selected peptides and therapeutic protein products [13] that can qualify as biosimilar candidates. As of August 2022, only 14 molecules in the EU and nine in the US have arrived as biosimilars.

## 11. Conclusions

After 17 years of using biosimilars and billions of doses administered with no safety or efficacy issues reported, it is about time for a rational rethinking of the regulatory procedure for their approval. Waivers of animal and human testing are now possible, based on 3D structural similarity being proven using newly discovered AI-based AlphaFold-2 testing. Biosimilar candidates with the exact same primary structures and highly similar analytics based on the newest technologies should be granted regulatory approval if it is proven that multiple AI simulations report the exact same 3D structures. The paradigm shift of making biosimilars accessible is now possible as over 160 prominent biosimilar candidates await entry into the market. If the theory presented in this paper appears radical, then the purpose of this communication is served; all paradigm shifts are radical.

## Figures and Tables

**Table 2 ijms-23-10690-t002:** Advanced analytical methods used in the analytical assessment of biosimilars.

Primary Structure
Intact Subunit Mass: ESI-Native MS; ESI-QTOF-MS; ESI-TOF-MS; GC-MS; HILIC-FLD, rCE-SDS; HILIC-FLD, RP-MS; LC-ESI-HRMS; LC-ESI-MS; LC-ESI-QTOF-MS; LC-ESI-TOF-MS; LC-ESI-Triple TOF-MS; LC-MS; LC-Orbitrap MS with CID/ETD; LC-QTOF-MS; LC-UV/MS; LC-MS; MALDI-TOF-MS; MALDI-TOF-MS; MALDI-TOF/RP-ESI-MS; MALDI-TOF-MS; RP-UPLC-Triple TOF MS; RP-Orbitrap MS; RP-ESI-HDMS; RP-ESI-MS; RP-ESI-MS with α-sialidase; RP-ESI-QTOF-MS; RP-ESI-TOF-MS; RP-Q Exactive-MS; RP-QTOF-MS; RP-UPLC- MSE; RP-UPLC-ESI-MS; RP-UPLC-ESI-QTOF-MS; RP-UPLC-QTOF-MS; RP-UPLC-QTOF-MS with lock spray ion source; RP-UV; RP-UV/ESI-MS; RP-UV/FLD/Exactive MS with HCD; RP-UV/QTOF-MS; RP-MS; RP/SEC-Triple TOF-MS; SEC-ESI-TOF-MS; SEC-UPLC UV/HESI-[Native] MS; SELDI-MS; UPLC-QTOF-MS; UPLC-ESI-MS; UPLC-HDMS; UPLC-MS
Peptide Mass: Acid hydrolysis method + RP-FLD; Edman degradation; ESI-TOF-MS; LC- MSE; LC-ESI- MSE; LC-ESI-QTOF-MS; LC-ESI/QQQ-MS; LC-hybrid ion trap-Orbitrap MS; LC-nanospray ion source-Orbitrap-MS with CID and ETD; LC-Orbitrap MS with CID/ETD/HCD; LC-QTOF-MS; LC-Triple TOF-MS; LC-UV/ESI-MS; LC-UV/MSE; LC-UV/Triple TOF-MS; MALDI-QTOF-MS; MALDI-TOF-MS; MS; nanoLC-ESI- MSE with CID; nanoLC-ESI-Orbitrap Fusion Tribrid MS with HCD; nanoLC-Ion Trap-Orbitrap with CID and ETD; nanoUPLC- MS; Q-Exactive MS; RP- UHR-UV/ESI-QTOF-MS; RP-ESI-Ion Trap MS; RP-ESI-MS; RP-ESI-TOF-MS; RP-QTOF-MS; RP-UHD-QTOF-MS; RP-UPLC- MSE; RP-UPLC-ESI-MS; RP-UPLC-Q Exactive-Orbitrap MS; RP-UPLC-QTOF-MS; RP-UPLC-QTOF-MS with lockspray ion source; RP-UPLC-UV-MSE; RP-UPLC-UV/ESI-MS; RP-UPLC-UV/ESI-Triple TOF-MS; RP-UPLC-UV/MS; RP-UV; RP-UV-QTOF-MS; RP-UV/ESI-MS; RP-UV/ESI-QTOF-MS; RP-UV/ESI-Triple TOF-MS; RP-UV/FLD; RP-UV/hybrid Ion Trap-Orbitrap MS; RP-UV/MS; RP-UV/Orbitrap-MS; RP-UV/Q Exactive MS; RP-UV/QTOF-MS; RP/MALDI-TOF-MS; UPLC-ESI-Hybrid MS; UPLC-HDMS; UPLC-QTOF- UV/MS; UPLC-QTOF-M
Orthogonal: HPLC; LC-MS; CD-MS; tandem MS (intact and subunit level); N-/C- terminal sequencing; microarray LC-MS; HILIC-FLD; rCE-SDS; Coefficient of determination; cosine of the angle; Bray–Curtis distance and nearness index; RP-MS; HILIC-FLD; RP-MS; LC-MS
**Higher Oder Structure**
Disulfide bridge/free -SH: Ellman assay; Free thiol FLD; LC-ESI-MS; LC-ESI-Triple TOF-MS; LC-hybrid ion trap-Orbitrap MS; LC-nano spray ion source-Orbitrap-MS with CID and ETD; LC-Orbitrap MS with CID/ETD/HCD; LC-UV/MSE; LC-UV/Triple TOF-MS; MALDI-TOF-MS; Measure-iT thiol assay; RP-ESI-MSE; RP-UPLC- MSE; RP-UPLC-ESI-Triple TOF-MS; RP-UPLC-Q Exactive-Orbitrap MS; RP-UPLC-QTOF-MS; RP-UPLC-UV-MSE; RP-UV/ESI-QTOF-MS; RP-UV/hybrid Ion Trap-Orbitrap MS; RP-UV/Q Exactive MS; RP-UV/QTOF-MS; RP/MALDI-TOF-MS; UPLC-ESI-Hybrid MS; UPLC-QTOF MS; UPLC-QTOF- UV/MS;
Secondary: Far UV CD; FTIR;
Tertiary: Near UV CD; 15N-HSQC) NMR; 1D NMR; 1H-NOESY); 1H-TOCSY) NMR; 2D (1H-13C-HSQC) NMR; 2D (1H-15N-HMQC) NMR; 2D (1H-15N-HSQC) NMR; 9G8A antibody binding assay; Antibody conformational array; ESI-IM-MS; FLR; HDX-MS; IM-MS; NanoESI-time-resolved HDX-MS; Near UV CD; NMR; QIM-TOF-MS; RP-FLD; UPLC-IM-MS; UV spectroscopy; XRC
Conformational stability: CIU with IM-MS; DSC; ITC; NanoDSC; TCSPC; VT-CD
Orthogonal: CD; FTIR; NMR; HDX-MS; SEC; thermal shift assay; NMR; SEC; AUC; DLS; CD; FLD; DSC; Far-UV; CD; FTIR; Raman spectroscopy; Near-UV; CD; fluorescence; DSC; DSF; NMR; X-ray crystallography; HDX-MS; AUC; crypto electron microscopy (EM)
**Glycosylation**
Glycopeptide: ESI-QTOF-MS; LC-ESI- MSE; LC-ESI-MS; LC-ESI-QTOF-MS; LC-MS; LC-nano spray ion source-Orbitrap-MS with CID and ETD; LC-UV/MSE; LC–ESI-MS; MALDI-TOF-MS; Q-Exactive MS; RP-HPLC-UV/MS with PNGase F; RP-UPLC- MSE with PNGase F + Asp-N; RP-UPLC-QTOF-MS; UPLC-QTOF-MS
Monosaccharide/sialic acid: Acid hydrolysis; Acid hydrolysis +RP-FLD; HPAEC-PAD; HPLC-FLD; IEX-UV; LC-UV/FLD; RP with fluorescein; RP-FLD; RP-FLD with DMB; Sialo oligosaccharide purified using SEC; UPLC-FLD with DMB; Weak AEX
Orthogonal: HILIC; Free glycan analysis; HILIC-FLD; RPLC-MS; DNPH; GRP derivatization
**Product-related attributes**
Aggregates/fragments: SEC-UV; 2D-PAGE; 90°LS; AF4; AUC-SE; AUC-SV; AUC-SV/SE; CE-SDS; CE-SDS-LIF; Congo red binding assay; DLS; FFF; FFF-LS; Gel Electrophoretic method with FLD; MALS; nrCE-SDS; Quantitative gel electrophoresis using TapeStation; SDS-PAGE; SEC-UPLC UV/HESI-[Native] MS; SEC-UV-MALS; SLS; SV-AUC; TEM; THT binding assay
Visible/Sub-visible particles: LM; LO; LO HIAC; MFI; NRM; NTA; Turbidimetry at 350 nm; URT
Charge variants: 2D-DIGE; 2D-PAGE; 2D-SDS-PAGE; AEX; CEX; cIEF; CZE; iCE; icIEF\IEF; IEX; SDS-PAGE
Orthogonal: CEX; CEX-MS; cIEF; CEX; HILIC-ESI-MS-CID-MS/MS
**Process-related attributes**
Related proteins: BAC; BAC-FLD; CE-SDS; HIC; HIC-FLD; LC-ESI-MS; LC-ESI-QTOF-MS; LC-MS; Non-denaturing RP; rCE-SDS; RP; RP-UV/FLD/MS; RP-UV/MS; SDS-PAGE; UPLC-ESI-MS; rCE-SDS
HCP: 2D LC (RPXRP)-QTOF-MS; 2D PAGE; 2D-LC-MSE; ELISA\UPLC-HDMS
HCD: Picogreen assay; qPCR; Threshold assay

Key: AEX/CEX: anion/cation exchange chromatography; AF4: asymmetrical field flow fractionation; ATR-FTIR: attenuated total reflection-Fourier transform infrared spectroscopy; AUC: analytical ultracentrifugation; BAC: boronate affinity chromatography; BLI: bioanalytical interferometry; CD: circular dichroism spectroscopy; CE: capillary electrophoresis; CGE: capillary gel electrophoresis; CID: collision-induced dissociation; cIEF: capillary isoelectric focusing; CIU: collision-induced unfolding; CSD: comparative signature diagrams; CZE: capillary zone electrophoresis; DLS: dynamic light scattering; DOSY: diffusion ordered spectroscopy; DSC: differential scanning calorimetry; DSF: differential scanning fluorimetry; ED: equilibrium dialysis; ELISA: enzyme-linked immunosorbent assay; ELISA: enzyme-linked immunosorbent assay; ESI: electrospray ionization; ETD: electron-transfer dissociation; FFF: field flow fractionation; FLD: fluorescence detection; FTICR: Fourier transform ion cyclotron resonance; FTIR: Fourier-transform infrared spectroscopy; GC: gas chromatography; HCD: higher-energy C-trap dissociation; HCD: host cell DNA; HCP: host cell protein; HDMS/HRMS: high definition/high resolution-mass spectrometry; HDX-MS: hydrogen-deuterium exchange-mass spectrometry; HDX: hydrogen-deuterium exchange; HESI: heated electrospray ionization; HIC: hydrophobic interaction chromatography; HILIC: hydrophilic interaction chromatography; HMQC: heteronuclear multiple quantum coherence; HMWs: high molecular weight species; HOS: higher-order structure; HPAEC: high-performance anion-exchange chromatography; HPLC: high-performance liquid chromatography; HSQC: heteronuclear single quantum coherence; ICD: isothermal chemical denaturation; iCE: imaged capillary electrophoresis; icIEF: imaged capillary isoelectric focusing; IdeS: immunoglobulin G-degrading enzyme of Streptococcus pyogenes; IEF: isoelectric focusing; IEX: ion exchange chromatography; IM-MS: ion mobility-mass spectrometry; IP-RP-AIF-IM-MS: ion pair-reversed phase-all ion fragmentation-ion mobility-mass spectrometry; IT-FLR: intrinsic fluorescence spectroscopy; ITC: isothermal titration calorimetry; LC: liquid chromatography; LIF: laser-induced fluorescence detection; LM: light microscopy; LO HIAC: light obscuration in high accuracy liquid particle counter; MALDI: matrix assisted laser desorption/ionization; MALS: multi-angle light scattering; MAM: multi-attribute methods; MFDS: Ministry of Food and Drug Safety; MFI: micro-flow imaging; ML: machine learning; MRM: multiple reaction monitoring; MS: mass spectrometry; MSE: tandem mass spectrometry; MST: microscale thermophoresis; nanoDSF: nano differential scanning fluorimetry; NMR: nuclear magnetic resonance; NMR: nuclear magnetic resonance spectroscopy; NOESY: nuclear Overhauser effect spectroscopy; NP: normal phase chromatography; nr/rCE-SDS: non-reduced/reduced capillary electrophoresis sodium dodecyl sulfate; NRM: Nile red microscopy; NTA: nanoparticle tracking analysis; PAD: pulsed amperometric detection; PAGE: polyacrylamide gel electrophoresis; PGSTE: pulsed-field gradient stimulated echo; PSA: pressure shift assay; PTMs: post-translational modifications; QIM-MS: quadruple ion neutral-mass spectrometry; qPCR: real-time/quantitative polymerase chain reaction; QQQ: triple quadrupole; QTOF: quadrupole time-of-flight; RP: reverse phase chromatography; SAXS: small angle X-ray scattering; SCX: strong cation exchange chromatography; SDS-PAGE: sodium dodecyl sulfate-polyacrylamide gel electrophoresis; SE/SV-AUC: sedimentation equilibrium/sedimentation velocity-analytical ultracentrifugation; SEC: size exclusion chromatography; SELDI: surface-enhanced laser desorption/ionization; SILAC: stable isotope labeling by/with amino acids in cell culture; SLS: static light scattering; SPR: surface plasmon resonance; TEM: transmission electron microscopy; TOCSY: total correlation spectroscopy; TOF: time-of-flight; TSA: thermal shift assay; UHD/UHR: ultra-high definition/ultra-high resolution; UPLC: ultra-performance liquid chromatography; URT: ultrasound resonance technology; UV: ultraviolet; UVPD: ultraviolet photodissociation; VEGF: vascular endothelial growth factor; VT-CD: variable temperature-circular dichroism; WAX: weak anion exchange chromatography; WCID: whole column imaging detection; XRC: X-ray crystallography; XRD: X-ray diffraction.

**Table 3 ijms-23-10690-t003:** Recombinant products with the expiry of the gene patents.

1: Albumin [1990-11-16]; 2: Insulin [2004-03-24]; 3: Interferon beta-1b [2004-03-28]; 4: Parathyroid hormone [2004-04-21]; 5: Interferon alfa-2b [2004-12-28]; 6: Plasma protein fraction [2005-01-19]; 7: Natalizumab [2005-01-30]; 8: Asparaginase [2005-03-08]; 9: Filgrastim [2005-08-23]; 10: Rho d immune globulin [2006-11-25]; 11: Gonadotropin, chorionic [2007-01-30]; 12: Sebelipase alfa [2007-04-17]; 13: Anakinra [2008-12-24]; 14: Alemtuzumab [2010-05-11]; 15: Abciximab [2010-09-14]; 16: Interferon alfa-n3 [2011-03-01]; 17: Antihemophilic factor/von willebrand factor complex [2011-07-19]; 18: Aldesleukin [2012-02-03]; 19: Etanercept [2012-03-07]; 20: Epoetin alfa [2012-08-15]; 21: Thrombin [2012-09-09]; 22: Ibritumomab tiuxetan [2012-11-13]; 23: Pegfilgrastim [2013-12-03]; 24: Bevacizumab [2014-01-25]; 25: Reslizumab [2014-06-17]; 26: Cetuximab [2014-08-10]; 27: Trastuzumab [2014-08-10]; 28: Trastuzumab hyaluronidase-oysk [2014-08-10]; 29: Pegaspargase [2014-10-20]; 30: Von willebrand factor [2014-11-14 ]; 31: Denileukin diftitox [2015-02-01]; 32: Rituximab [2015-02-01]; 33: Hyaluronidase [2015-02-01]; 34: Sargramostim [2015-02-01]; 35: Imiglucerase [2015-03-24]; 36: Menotropins (fsh, lh) [2015-09-29]; 37: Palifermin [2015-09-29]; 38: Urofollitropin [2015-09-29]; 39: Peginterferon alfa-2b [2015-11-02]; 40: Basiliximab [2016-05-16]; 41: Daclizumab [2016-05-16]; 42: Alteplase [2016-05-22]; 43: Equine thymocyte immune globulin [2016-09-05]; 44: Denosumab [2016-12-23]; 45: Insulin lispro [2017-01-10]; 46: Infliximab [2017-04-04]; 47: Panitumumab [2017-05-05]; 48: Adalimumab [2017-09-26]; 49: Blinatumomab [2018-04-21]; 50: Insulin aspart [2018-05-19]; 51: Rilonacept [2018-09-25]; 52: Romiplostim [2018-10-23]; 53: plasma proteins [2018-12-10]; 54: Interferon beta-1a [2020-01-14]; 55: Interferon gamma-1b [2020-01-14]; 56: Palivizumab [2020-05-03]; 57: Capromab pendetide [2020-08-21]; 58: Ranibizumab [2020-08-24]; 59: Gemtuzumab ozogamicin [2020-11-28]; 60: Ocriplasmin [2020-12-21]; 61: Abatacept [2021-02-15]; 62: Golimumab [2021-03-07]; 63: Rasburicase [2021-05-01]; 64: Dornase alfa [2021-09-04]; 65: Tenecteplase [2021-10-30]; 66: Insulin detemir [2021-11-19]; 67: Laronidase [2021-11-30]; 68: Follitropin alfa/beta [2022-01-22]; 69: Pertuzumab [2022-05-17]; 70: Peginterferon alfa-2a [2022-08-01]; 71: Corticorelin ovine triflutate [2022-08-05]; 72: Omalizumab [2022-08-14]; 73: Darbepoetin alfa [2022-08-29]; 74: Pegvisomant [2023-01-09]; 75: Imciromab pentetate [2023-03-05]; 76: Thyrotropin alfa [2023-06-24]; 77: Agalsidase beta [2023-10-01]; 78: Selumetinib [2023-12-12]; 79: Albiglutide [2025-01-04]; 80: Antihemophilic factor [2025-01-04]; 81: Insulin aspart [2025-01-04]; 82: Lixisenatide [2025-01-04]; 83: Methoxy polyethylene glycol-epoetin beta [2025-01-04]; 84: Aflibercept [2025-03-15]; 85: Ramucirumab [2025-03-15]; 86: Belimumab [2025-05-20]; 87: Insulin glulisine [2025-05-23]; 88: Certolizumab pegol [2025-11-08]; 89: Ipilimumab [2025-11-08]; 90: Fremanezumab-vfrm [2025-11-14]; 91: Calfactant [2026-01-10]; 92: Tocilizumab [2026-04-13]; 93: Eculizumab [2026-04-27]; 94: Inotuzumab ozogamicin [2026-06-23]; 95: Mepolizumab [2026-06-23]; 96: Ocrelizumab [2026-06-23]; 97: Ofatumumab [2026-06-23]; 98: Raxibacumab [2026-06-23]; 99: Coagulation factor ix [2026-07-13]; 100: Desirudin [2026-08-02]; 101: Poractant alfa [2026-11-02]; 102: Benralizumab [2027-01-11]; 103: Galsulfase [2027-06-13]; 104: Evolocumab [2027-08-23]; 105: Ustekinumab [2027-11-30]; 106: Capmatinib [2027-12-12]; 107: Ado-trastuzumab emtansine [2028-10-22]; 108: Belatacept [2028-12-05]; 109: Canakinumab [2030-03-29]; 110: Abobotulinumtoxina [2030-03-30]; 111: Asparaginase erwinia chrysanthemi [2030-03-30]; 112: Incobotulinumtoxina [2030-03-30]; 113: Dulaglutide [2030-05-05]; 114: Metreleptin [2030-05-05]; 115: Brentuximab vedotin [2030-05-26]; 116: Insulin degludec [2030-06-24]; 117: Necitumumab [2030-09-08]; 118: Brodalumab [2030-10-08]; 119: Secukinumab [2030-10-08]; 120: Nivolumab [2031-02-04]; 121: Pembrolizumab [2031-02-04]; 122: Insulin detemir [2031-04-01]; 123: Vedolizumab [2031-05-02]; 124: Elotuzumab [2031-08-05]; 125: Olaratumab [2031-08-05]; 126: Siltuximab [2031-08-05]; 127: Sarilumab [2031-10-11]; 128: Alirocumab [2031-10-25]; 129: Alirocumab [2031-10-25]; 130: Insulin degludec [2031-10-25]; 131: Liraglutide [2031-10-25]; 132: Insulin glargine [2031-10-25]; 133: Insulin lispro [2031-10-25]; 134: Ixekizumab [2031-10-25]; 135: Obinutuzumab [2031-10-25]; 136: Peginterferon beta-1a [2031-10-25]; 137: Daratumumab [2031-10-28]; 138: Somatropin [2031-12-05]; 139: Eptinezumab-jjmr [2031-12-12]; 140: Atezolizumab [2032-08-13]; 141: Avelumab [2032-08-14]; 142: Durvalumab [2032-08-14]; 143: Ziv-aflibercept [2032-10-31]; 144: Choriogonadotropin alfa [2033-02-06]; 145: Alglucosidase alfa [2033-03-11]; 146: Insulin susp isophane [2033-03-11]; 147: Sacrosidase [2033-03-11]; 148: Dupilumab [2033-03-14]; 149: Coagulation factor viia [2033-04-24]; 150: Bezlotoxumab [2033-09-09]; 151: Pemigatinib [2033-12-12]; 152: Sacituzumab govitecan-hziy [2033-12-12]; 153: Tucatinib [2033-12-12]; 154: Ravulizumab-cwvz [2034-03-07]; 155: Idarucizumab [2034-07-31]; 156: Dinutuximab [2034-10-06]; 157: Guselkumab [2035-02-24]; 158: Obiltoxaximab [2035-02-24]; 159: Isatuximab-irfc [2035-10-05]; 160: Taliglucerase alfa [2036-02-11]; 161: Velaglucerase alfa [2036-02-11]; 162: Polatuzumab vedotin-piiq [2039-10-22].

## Data Availability

Not Applicable.

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
