# Peer review of "Molecular Biosimilarity—An AI-Driven Paradigm Shift"

_ijms, 2022, doi:10.3390/ijms231810690_

Round 1

Reviewer 1 Report

In this paper the author suggests "paradigm shift" in approval of biosimilars, suggesting establishing analytics-based molecular biosimilarity and wiaving in vivo testing.

The author excessively and unnecessarily explains the procedure for accepting biosimilars (In analytical assessment the author listed the standard methods for testing used in the analytical assessment of biosimilars which are well known), suggesting that animal testing, clinical pharmacology, imunogenecity and clinical efficiancy should be waived in biosimilar approvals.

I agree that the science of analytical testing has progressed to a point where we can now clearly point out even the most minor differences in molecular structures. However, I find that this paper is engaged in order to exert pressure on organizations that pass regulations, but without clear evidences, especially in the field of immunogenicity.

 I condsider this a political issue rather than scientific and personally I don’t like this paper but I think it can be published.

Author Response

I am extremely obliged to the reviewer for sharing a frank comment. And I agree that this is pressure on regulatory agencies to reconsider their views, and if it is political pressure, I am willing to accept it. One agency, MHRA, has already done it, though not as suggested. My argument is simple; if the two molecular structures are the same (as same as they can be), then there is little need to do any additional tests. immunogenicity is also a function of the structure. The new AI allows us to conclude that the 3D will always be the same for some molecules tested repeatedly in a dry lab using AI. The "abundance of caution" has kept the entry of biosimilars so difficult due to the cost of compliance, which I have shown that it never fails. In my humble opinion, an article should stir up debate, and if conservative minds can be stirred up, it may lead to great advances in the science of biosimilars. Would I succeed, I do not know, but if I had not tired, I would never have. So, thanks again for your frankness, and I hope to stir up the dust far and wide if you allow its publication. While not recommended, I am submitting the manuscript for professional editing by the MDPI before resubmitting.

Reviewer 2 Report

This manuscript aimed to describe the new approach in the process of the approval of biosimilars. Overall, this manuscript is comprehensive, provides much useful information and it brings some novelty in the research field. However, my main concern is regarding the type of the article. In my opinion, this should be an ‘Opinion article’ since it provides critical assessment of the topic of biosimilar approval, which many researchers may oppose. According to the journal instructions, review articles should provide a complete and balanced overview of the latest progress in a given area of research. In my opinion, the author provided evidence for this simplistic approach to biosimilars approval that includes only analytics-based molecular biosimilarity evaluation, but the opposing evidence is lacking in some parts.

In accordance with that, many paragraphs of the manuscript do not have references. There are 90 references in the whole manuscript on 25 pages, which confirms that a large part of the text is an ‘opinion’ and not an ‘overview’.

There are few technical concerns in the manuscript as well:

- the terms ‘in vitro’ and ‘in vivo’ should be in italic;

- line 9: ‘safe safety’;

- abbreviations AI and MS should be defined in the abstract, despite they might be obvious;

- line 43: ‘in eukaryotes like humans and Chinese Hamster Ovary cells’;

- some parts of the text I find redundant, such as: ‘A chain of amino acids is created by connecting the carboxyl group of one amino acid with the amino group…’ (lines 52-53); there are some other similar examples as well.

Author Response

I am very thankful to the reviewer for pointing out the mistakes; these should have been corrected before I submitted the manuscript. I took care of all identified and similar issues in the manuscript. I would have no issue if this article is listed as "opinion" instead; however, it does include new information about the AI methodologies, details of the current regulatory status of approval of biosimilars, and presents an analytical perspective. I will leave this to the publisher to decide. I have sent out my revised manuscript for professional editing by the MDPI service to make sure it has fewer mistakes.

Round 2

Reviewer 2 Report

Corrections according to my comments have been done.